# Perioperative Factors Impact on Mortality and Survival Rate of Geriatric Patients Undergoing Surgery in the COVID-19 Pandemic: A Prospective Cohort Study in Indonesia

**DOI:** 10.3390/jcm11185292

**Published:** 2022-09-08

**Authors:** Nancy Margarita Rehatta, Susilo Chandra, Djayanti Sari, Mayang Indah Lestari, Tjokorda Gde Agung Senapathi, Haizah Nurdin, Belindo Wirabuana, Bintang Pramodana, Adinda Putra Pradhana, Isngadi Isngadi, Novita Anggraeni, Kenanga Marwan Sikumbang, Radian Ahmad Halimi, Zafrullah Khany Jasa, Akhyar Hamonangan Nasution, Mochamat Mochamat, Purwoko Purwoko

**Affiliations:** 1Anesthesiology and Reanimation, Faculty of Medicine, Airlangga University, Surabaya 60131, East Java, Indonesia; 2Anesthesiology and Intensive Care, Faculty of Medicine, University of Indonesia, Jakarta City 10430, Jakarta, Indonesia; 3Anesthesiology and Intensive Therapy, Faculty of Medicine Public Health and Nursing, Gadjah Mada University, Sleman 55281, Yogyakarta, Indonesia; 4Anesthesiology and Intensive Care, Faculty of Medicine, Sriwijaya University, Palembang 30126, South Sumatera, Indonesia; 5Anesthesiology and Intensive Care, Faculty of Medicine, Udayana University, Denpasar 80361, Bali, Indonesia; 6Anesthesiology, Intensive Therapy and Pain Management, Faculty of Medicine, Hasanuddin University, Makassar 90245, South Sulawesi, Indonesia; 7Anesthesiology and Reanimation, Faculty of Medicine, Brawijaya University, Malang 65125, East Java, Indonesia; 8Anesthesiology and Intensive Care, Faculty of Medicine, University of Riau, Pekanbaru 28133, Riau, Indonesia; 9Anesthesiology and Intensive Care, Faculty of Medicine, Lambung Mangkurat University, Banjarmasin 70233, South Kalimantan, Indonesia; 10Anesthesiology and Intensive Care, Faculty of Medicine, University of Padjadjaran, Bandung 45363, West Java, Indonesia; 11Anesthesiology and Intensive Therapy, Faculty of Medicine, Syiah Kuala University, Banda Aceh 24415, Aceh, Indonesia; 12Anesthesiology and Intensive Care, Faculty of Medicine, University of Sumatera Utara, Medan 20136, North Sumatera, Indonesia; 13Anesthesiology and Intensive Care, Faculty of Medicine, Diponegoro University, Semarang 50275, Central Java, Indonesia; 14Anesthesiology and Intensive Care, Faculty of Medicine, Sebelas Maret University, Surakarta 57126, Central Java, Indonesia

**Keywords:** geriatric anesthesia, COVID-19, perioperative factors, mortality, survival rate

## Abstract

**Background:** The COVID-19 pandemic continues to have an impact on geriatric patients worldwide since geriatrics itself is an age group with a high risk due to declined physiological function and many comorbidities, especially for those who undergo surgery. In this study, we determine the association between perioperative factors with 30-day mortality and a survival rate of geriatric patients undergoing surgery during COVID-19 pandemic. **Methods:** A prospective cohort study was conducted at 14 central hospitals in Indonesia. The recorded variables were perioperative factors, 30-day mortality, and survival rate. Analyses of associations between variables and 30-day mortality were performed using univariate/multivariable logistic regression, and survival rates were determined with Kaplan–Meier survival analysis. **Results:** We analyzed 1621 elderly patients. The total number of patients who survived within 30 days of observation was 4.3%. Several perioperative factors were associated with 30-day mortality (*p* < 0.05) is COVID-19 (OR, 4.34; 95% CI, 1.04–18.07; *p* = 0.04), CCI > 3 ( odds ratio [OR], 2.33; 95% confidence interval [CI], 1.03–5.26; *p* = 0.04), emergency surgery (OR, 3.70; 95% CI, 1.96–7.00; *p* ≤ 0.01), postoperative ICU care (OR, 2.70; 95% CI, 1.32–5.53; *p* = 0.01), and adverse events (AEs) in the ICU (OR, 3.43; 95% CI, 1.32–8.96; *p* = 0.01). Aligned with these findings, COVID-19, CCI > 3, and comorbidities have a log-rank *p* < 0.05. The six comorbidities that have log-rank *p* < 0.05 are moderate-to-severe renal disease (log-rank *p* ≤ 0.01), cerebrovascular disease (log-rank *p* ≤ 0.01), diabetes with chronic complications (log-rank *p* = 0.03), metastatic solid tumor (log-rank *p* = 0.02), dementia (log-rank *p* ≤ 0.01), and rheumatology disease (log-rank *p* = 0.03). **Conclusions:** Having at least one of these conditions, such as COVID-19, comorbidities, emergency surgery, postoperative ICU care, or an AE in the ICU were associated with increased mortality in geriatric patients undergoing surgery during the COVID-19 pandemic.

## 1. Introduction

The COVID-19 pandemic has created a healthcare crisis in the world. Due to a large number of COVID-19 cases, many hospitals shifted to serve COVID-19 patients compared to those without. Consequently, the number of surgeries performed dropped globally by about 25 million elective surgeries in early 2020 [1]. This also had an impact on geriatric patients’ surgical services. Geriatric patients have decreased physiological function (e.g., diminished cognitive function, decreased cardiac output, increased mean arterial pressure, decreased arterial oxyhemoglobin, and lowered glomerular filtration rate) and have many comorbidities (e.g., cardiovascular disease, diabetes, chronic respiratory disease, or cancer). This alone puts geriatric patients at risk of undergoing surgery [2]. Undergoing surgery during the COVID-19 pandemic also has a high risk. Patients may not be infected with COVID-19 at the initial admission, but the patients may be exposed during treatment. Having COVID-19 may increase mortality and decreased survival rate. The combination of those worsens the outcome of surgical and anesthesia services in geriatric patients [1,3].

The research included 1128 patients from 235 hospitals in Europe, Africa, Asia, and North America, showing high mortality rate (23.8%) in patients COVID-19 who underwent surgery. The mortality rate of patients older than 70 years and infected with COVID-19 is high compared to people at a young age, 33.7% vs. 13.9% [4]. COVID-19 has been shown to increase the Intensive Care Unit (ICU) admissions, length of stay patients in hospital, and mortality, and decrease the survival rate patients [5,6].

Many factors affect the mortality and survival of patients undergoing surgery [7], but not much is known about the factors that affect mortality and survival during COVID-19 pandemic, especially in geriatric patients. Thus, it is important to know how perioperative factors impact the mortality and survival rates of geriatric patients undergoing surgery in the COVID-19 pandemic, since it may influence the decision-making and identification of patients who benefit from surgery requiring anesthesia.

## 2. Materials and Methods

### 2.1. Study Design

This study is a prospective cohort study. This research was conducted in 14 teaching hospitals for Anesthesiology and Intensive Therapy Residency Education in Indonesia. The data collection was conducted for four months, including three months of data collection from February to April 2021, and we prospectively followed up for 30 days. Every collector was briefed and had standardized training before the data collection. We enrolled patients after their initial consultation with the anesthesiologist.

### 2.2. Sample and Population

We recruited patients who underwent surgery and required anesthesia. We included all patients aged 60 years or older who underwent surgery from all disciplines between February and April 2021. Patients willing to participate in this study with complete records of medical comorbidities were included. We excluded all patients who did not provide consent and did not complete the follow-up within the established follow-up periods.

### 2.3. Parameters

The variables included preoperative factors, including age, gender, BMI, comorbidities based on the Charlson Comorbidity Index (CCI) score, COVID-19 status, ASA score, nonemergency and emergency surgery, and anesthesia techniques. The patients’ ages were categorized into two groups: 60–84 and ≥85 [8]. The BMI was divided into two groups: normal (18.5–24.9) and under/overweight (<18.5 or ≥25 kg/m^2^). The CCI score is a scoring system that is used to predict prognosis based on the comorbidities of patients. It was divided into two categories: CCI ≤ 3 and CCI > 3 [9]. The status of COVID-19 was determined using a preoperative nasopharyngeal swab (Rapid antigen or PCR SARS-CoV-2) within the period of one day before the day of surgery. The anesthesia technique was divided into two groups: nongeneral anesthesia and general anesthesia. These variables were carried out in preanesthesia assessment.

Intraoperative factors include adverse events (AEs) that occur in the operating room, such as hypotension, hemorrhage more than estimated blood volume (EBV), prolonged surgery time, oxygen saturation, anesthesia awareness, anesthesia technique conversion, and prolonged block. Postoperative factors included AE in the recovery room, postoperative ICU care, AE in the ICU, and AE inward. AEs in the recovery room were hypotension, shivering, delayed emergence, hemorrhage more than EBV, postoperative nausea and vomiting (PONV), pain, oxygen desaturation, and hypothermia. AEs in the ICU were prolonged length of stay in ICU, unexpected ventilation used, and nosocomial infection. AE inward are pain, hypotension, PONV, hemorrhage, and oxygen desaturation. Patients were admitted to the ICU when the patient needed intensive monitoring, cardiovascular stabilization, unstable condition, or had a risk for severe complications after surgery [10,11]. These were recorded if patients have it during surgery or/and postoperatively. See the definition in Table 1.

### 2.4. Outcome Measures

The output variables in this study included mortality within 30 days after surgery and survival rate. The participants were followed prospectively for 30 days, whether the participants had died or not, and the survival time was recorded.

### 2.5. Statistical Analysis

The statistical tests were performed with the SPSS^®^ 23 software program (IBM Corp., Armonk, NY, USA). Numerical variables with a normal distribution are shown as the mean ± standard deviation (SD) or if not normal as the median (minimum-maximum). Missing data were excluded from the analyses. Categorical variables are displayed as numbers (percentages). Next, we summarized the data in the form of tables and narratives. We determined the association between perioperative factors and mortality using chi-square tests and, if *p* < 0.05, the analysis was subjected to multivariable logistic regression analysis. We used Kaplan–Meier survival analysis to describe the survival rate in COVID-19, comorbidities, and CCI. If *p* < 0.05, it is displayed in figure form.

### 2.6. Ethical Feasibility

Overall, this study was approved by the local ethics committees and institutional review boards, including the Medical and Health Research Ethics Committee (MHREC), Faculty of Medicine, Public Health, and Nursing, Universitas Gadjah Mada—Dr. Sardjito General Hospital with approval number KE/FK/1381/EC/2020. The ethics committee waived the written informed consent from all subjects, a legal surrogate, or the parent.

## 3. Results

A total of 2091 patients were included in this study, and 470 patients were excluded due to incomplete data. Finally, 1621 patients were analyzed, with 1551 patients who survived (95.7%) and 70 patients who died (4.3%). The average age of the patients was 67.1 ± 6.2 years, and most were male (54%) with almost all of the patients (98.9%) being below 84 years of age. On average, the majority of the patients (60%) had normal BMI, most of the patients (90%) underwent nonemergency surgery, and most had an ASA classification of I and II, with general anesthesia being the most commonly used technique. The majority of the patients (60%) had at least one comorbidity, with as many as 95.4% with CCI ≤ 3 and most of them (99.1%) were non-COVID. The most common comorbidity was hypertension, which was found in approximately 30% of the patients (see Table 2).

Based on the place where AE occurred, hypotension was observed in the intraoperative and recovery rooms (12.7% and 4.1%, respectively), a prolonged length of stay was observed in the ICU (17.2%), and pain was observed in the ward (3.7%). As many as 13% of the geriatric patients after surgery were admitted to the ICU (see Table 2).

Five variables had *p* < 0.05 with univariate/multivariable logistic regression analyses. These were COVID-19 (OR, 4.34; 95% CI, 1.04–18.07; *p* = 0.04), CCI > 3 (OR, 2.33; 95% CI, 1.03–5.26; *p* = 0.04), emergency surgery (OR, 3.70; 95% CI, 1.96–7.00; *p* ≤ 0.01), postoperative ICU care (OR, 2.70; 95% CI, 1.32–5.53; *p* = 0.01), and AE in the ICU (OR, 3.43; 95% CI, 1.32–8.96; *p* = 0.01) (see Table 3).

We also analyzed the survival rate with Kaplan–Meier survival analysis. The patients with COVID-19, comorbidities, CCI > 3, and any of six comorbidities had a decreasing survival rate (log-rank *p* < 0.05). In addition, the geriatric patients with COVID-19, comorbidities, and CCI > 3 also had a lower survival rate than those without (see Figure 1). The six comorbidities were a moderate-to-severe renal disease, cerebrovascular disease, diabetes with chronic complications, metastatic solid tumor, dementia, and rheumatologic disease (see Figure 2).

## 4. Discussion

There are challenges associated with administering anesthesia to aging patients. As the number of older people increases, more surgeries will be performed. Unfortunately, the problems faced in Indonesia are exacerbated by limited healthcare facilities, and these problems were worsened in the COVID-19 pandemic.

Turrentine et al. [12] showed that geriatric patients who underwent surgery had a 28% morbidity rate and 2.3% mortality rate, which increased in people older than 80 years of age. These morbidity and mortality rates were obtained pre-COVID-19 pandemic. In Indonesia, there are no data on the 30-day mortality rate of geriatric patients undergoing surgery in the time before the COVID-19 pandemic. Therefore, at Hasan Sadikin General Hospital in 2017–2019, the mortality rate was 0.2% for patients of all ages [7]. In addition, we found that 4.3% of geriatric patients died within 30 days after undergoing surgery during the COVID-19 pandemic. This 30-day mortality rate is quite low compared to that in Italy, where the mortality rate is 8.13% [13]. However, we found that the proportion of mortality rate in geriatric patients with COVID-19 was higher, 26.6%, than that in Italy, 19.51% [13]. This means that, during the COVID-19 pandemic, 30-day mortality was higher when compared to prepandemic. These findings are supported by our subsequent analysis.

In our study, we found that COVID-19, a CCI score > 3, emergency surgery, postoperative ICU care, an AE in the ICU was associated with 30-day mortality in geriatric patients who received anesthesia services. This finding supports that of other studies that were focused on geriatric patients. Several risk factors are associated with postoperative mortality in geriatric patients, such as older age, sex, comorbidities, higher ASA classification, anesthesia method, longer length of stay, complications in perioperative care, ICU admission, and COVID-19 infection [13,14]. In one study in Turkey, older age, emergency surgery, several preexisting comorbidities, a longer preoperative hospital stay, postoperative complications, ICU admission, and ventilator use were associated with increased mortality in geriatric patients undergoing orthopedic surgery [15].

Geriatric patients with COVID-19 have a 4.34 times greater risk than geriatric patients without COVID-19. This is because geriatric patients are more susceptible to adverse clinical outcomes related to COVID-19 infection, and treatment in geriatric patients is more challenging since they tend to face more comorbid conditions and side effects of polypharmacotherapy [16]. In addition, one systematic review showed that COVID-19 infection increases 30-day mortality, inpatient mortality, and ICU admission among elderly hip fracture patients [5].

The CCI is associated with all-cause mortality after discharge. A high CCI score reflects that patients have more comorbidities. Initially, the CCI was used to classify comorbid conditions in patients and determine the effects of comorbidities on mortality at 1 year and 10 years [17]. At present, the CCI is used to evaluating several outcomes related to surgery or intensive care settings [18,19]. The CCI may be used to predict surgical outcomes such as 30-day mortality, 30-day readmission, 1-year mortality, complications, and failure to rescue. A higher CCI score may mean the risk of mortality increases by 16.9% at 21 months after surgery [18]. In our study, there was an association between a CCI score > 3 and 30-day mortality (*p* = 0.04), but interestingly, there was no association found in geriatric patients who had at least one comorbidity or no comorbidities (*p* = 0.33). Our finding is in line with one study in Israel that showed that a higher CCI score increased 30-day postoperative mortality in all groups of geriatric patients [20]. In addition, our findings show that, even though geriatric patients have comorbidities, the determination of the prognosis related to mortality must be seen from the existing comorbid classification, such as the CCI score. Many researchers who have used the CCI to predict postoperative mortality in the short term have used an average or categorical CCI score instead of using a single score [20,21,22,23]. The CCI score is used not only to predict the mortality risk but also to predict a prolonged length of hospital stay, and it can predict a high number of complications that may occur or patient readmission after surgery [9,18].

We found that emergency surgery has a 3.70 times greater risk for mortality than nonemergency surgery. This is because emergency admissions to the hospital may often occur during out-of-office periods, weekends, or at night, and emergency surgery may involve the risk of not having a well-defined management plan. The important consideration for emergency patients is that the intervention may not improve the patient’s condition as well as elective surgery [8].

Geriatric patients have a 2.70 times greater risk of increased mortality with postoperative ICU care. Since some surgeries are high-risk, postoperative ICU care should involve contingency planning to achieve a better postsurgical outcome. It may be necessary to admit surgical patients to the ICU admission because of known preoperative risk factors or due to unpreventable intraoperative events. In those cases, it is necessary to admit patients to the ICU for a better outcome. However, ICU patients have a high mortality rate due to critical factors, such as their condition or intraoperative/postoperative AE [8]. Geriatric patients having an AE in the ICU increase the risk of mortality by 3.43 times. Patients in the ICU also have a high risk of treatment-related injury. Having an AE during ICU care may further increase the mortality rate [8]. The occurrence of an AE is associated with an increased potential for serious problems in intensive care, a higher rate of mortality, and a prolonged length of stay.

Several factors impacted the survival rate in our study. COVID-19, comorbidities, a CCI score > 3, and six comorbidities were associated with a decreased survival rate in the Kaplan-Meier analysis. These findings align with our regression analysis that indicated that COVID-19 and a CCI score > 3 have a significant association with mortality. The six comorbidities are moderate-to-severe renal disease, cerebrovascular disease, diabetes with chronic complications, metastatic solid tumor, dementia, and rheumatologic disease, which are associated with lower survival rates in geriatric patients who receive anesthesia. Our finding is in line with a study in Taiwan. This study shows that having at least one of these conditions, such as congestive heart failure, chronic obstructive pulmonary disease, diabetes, solid tumor metastasis, and a CCI score ≥ 3, decreases the survival rate in geriatric patients after hip surgery [24]. In addition, COVID-19 is an independent risk factor for a decreased survival rate. One of the retrospective cohort studies in Iran showed that COVID-19 decreased the survival rate in almost all patients of all ages; however, patients aged > 70 years old were the most affected group [6].

Moderate-to-severe renal disease is associated with an increased risk of death, increased cardiovascular events, and hospitalization, and it also increases the incidence of an AE after elective surgery [25]. Renal impairment in geriatric patients can lead to an increased burden of diseases, such as diabetes and hypertension, decreasing the clearance of renal excreted medication, increasing the risk of dehydration, and increasing electrolyte imbalance [26,27].

Cerebrovascular disease has the highest mortality rate and is also an important public health problem that leads to high rates of disability among elderly individuals [25]. Geriatric patients with cerebrovascular disease may have a decreased ability to perform daily living activities, an increased risk of postoperative cognitive dysfunction and delirium, and an increased sensitivity to neuraxial and regional anesthesia [26].

We found that patients with diabetes did not have a decreased survival rate, but chronic complications can decrease the survival rate. If patients have chronic complications caused by diabetes, it indicates that the patients have poor control of their blood glucose. Hyperglycemia can increase the risk of postoperative infection, leading to longer hospitalization stays, higher ICU admissions, and increased morbidity and mortality [28].

Patients with oncological disease are rarely referred for surgery due to poor outcomes and are directed to palliative care [29]. The incidence of cancer in geriatric patients is increased by 26% compared to that in younger aged patients, and cancer-related mortality increases by 15% in patients ≥ 65 years [30]. Geriatric patients who undergo surgery may have an impaired stress response, increased senescence, and decreased immunity after surgery. Therefore, we must consider the risks and benefits before planning to perform surgery in oncology patients, especially geriatric patients [23].

At present, dementia has been identified as a major challenge to health care worldwide. Anesthesia use can worsen dementia, especially Alzheimer’s disease. Volatile anesthetics such as halothane and isoflurane increase oligomerization and cytotoxicity of amyloid peptide, leading to worsening of the pathology of dementia. In addition, the use of sevoflurane and isoflurane can cause neuronal apoptosis and alpha-beta protein aggregation, increasing behavioral impairment and mortality [31].

Rheumatologic disease is a chronic inflammatory disease of unknown etiology. Rheumatic disorders have high variability; some develop very rapidly, while others develop chronically, causing disability. One of the rheumatologic diseases is rheumatoid arthritis. This disease is characterized by deformities, which may affect the patients’ positioning during surgery and make it difficult to access the area that requires anesthesia (especially regional anesthesia). In addition, head and neck involvement in rheumatoid arthritis can result in difficulties with impaired airways due to the complexity of several maneuvers that are necessary for tracheal intubation. Therefore, it is important to evaluate the extension of the cervical spine, temporomandibular joint, and cricoarytenoid joint involvement in preoperative assessments [32]. Patients with rheumatologic disease have a higher risk of myocardial infarction, which is similar to individuals with diabetes or a person ten years older than the age of the patients [33].

In our study, there are several limitations. First, not all the hospitals in Indonesia were involved. This study was conducted in 14 teaching hospitals which is a referral hospital, including referrals for COVID-19 patients where the cases encountered are complex. Second, although 30 days is enough to see mortality and survival rate of geriatric patients undergoing surgery, it may be necessary to see survival with longer follow-up. Third, we did not provide causes of death because death can occur from various causes and is not always directly related to the study. Fourth, AEs (such as hypoxia, hypotension, and hypothermia) were measured and recorded from the presence or absence during or after surgery and did not include more detailed data on the duration and frequency of occurrence. Lastly, although hemorrhage more than EBV was recorded as one of the AEs, we did not include blood transfusion as one of the perioperative factors that could affect the patient’s mortality and survival rate. We recommend that further research be conducted on blood transfusion as a factor that may be related to patients’ mortality and survival rate. Nevertheless, apart from the limitations mentioned above, our results may sufficiently represent the general geriatric patients who undergo surgery in Indonesia, especially during the COVID-19 pandemic.

## 5. Conclusions

The overall 30-day mortality rate of geriatric patients undergoing surgery was 4.3%. Five variables were associated with the 30-day mortality rate in geriatric patients undergoing surgery, including COVID-19 infection, a CCI score > 3, emergency surgery, postoperative ICU, and an AE in the ICU. Aligned with these findings, COVID-19, comorbidities, and a CCI score > 3 have lower survival rates.

Using a single comorbidity alone may not predict the 30-day mortality rate. Comorbidities became relevant when converted into categories using the Charlson Comorbidity Index score, which can be used to predict the 30-day mortality rate and the decrease in survival rate in geriatric patients. Based on these results, an anesthesiologist may be more aware of the perioperative factors that may affect the 30-day mortality rate in geriatric patients, such as COVID-19 infection, a CCI score > 3, undergoing emergency surgery, and admission to the ICU, especially if an AE occurs. In addition, when geriatric patients have one of these conditions, such as COVID-19 infection, a CCI score > 3, moderate-to-severe renal disease, cerebrovascular disease, diabetes with chronic complications, a metastatic solid tumor, dementia, or rheumatologic disease, it may decrease the survival rate of geriatric patients.

Finally, we realize that each country and each place have different perioperative risk factors for geriatric patients undergoing surgery. In addition to being useful for Indonesian local data, of course, this research can be useful when applied to populations in countries because COVID-19 affected many countries in terms of rapid transmission and high mortality rates. Additional information obtained in various countries (including our research) can improve our understanding, help us identify better treatment methods, prevent a worsening condition, and improve the prognosis of geriatric patients undergoing surgery during the COVID-19 pandemic

## Figures and Tables

**Figure 1 jcm-11-05292-f001:**
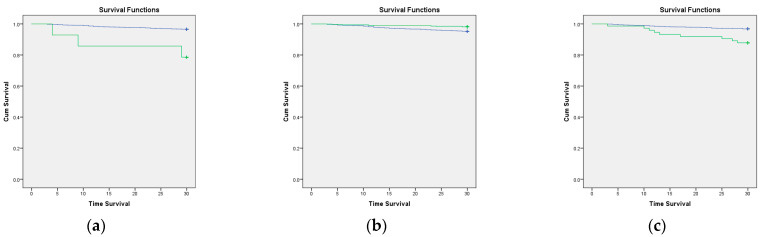
Kaplan–Meier curve of the 30-day survival rate according to COVID-19 status, CCI score, and comorbidities. The green line is patients who have that variable, and the blue line is patients without that variable. (**a**) COVID-19 status (log-rank *p* ≤ 0.01; (**b**) comorbidities (log-rank *p* ≤ 0.01); and (**c**) CCI > 3 (log-rank *p* ≤ 0.01).

**Figure 2 jcm-11-05292-f002:**
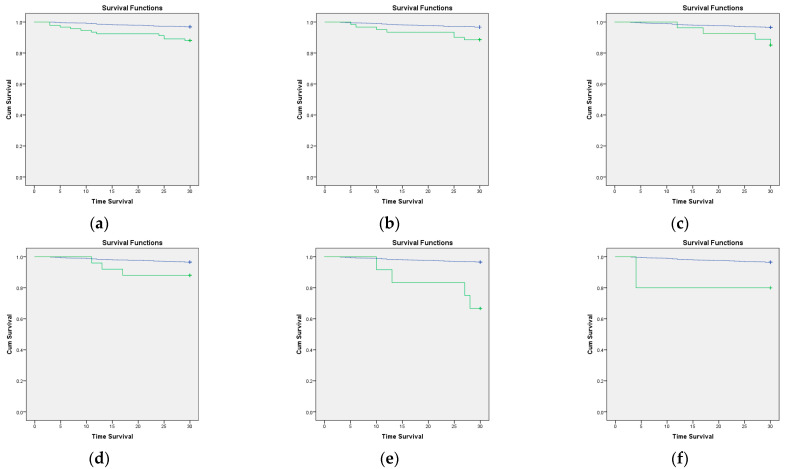
Kaplan–Meier curve of the 30-day survival rate of comorbidity in geriatric patients. The green line indicates patients with comorbidities, and the blue line indicates patients with no comorbidities. (**a**) Moderate-to-severe renal disease (log-rank *p* ≤ 0.01); (**b**) cerebrovascular disease (log-rank *p* ≤ 0.01); (**c**) diabetes with chronic complications (log-rank *p* = 0.03); (**d**) metastatic solid tumor (log-rank *p* = 0.02); (**e**) dementia (log-rank *p* ≤ 0.01), and (**f**) rheumatologic disease (log-rank *p* = 0.03).

**Table 1 jcm-11-05292-t001:** Parameter definition of variables.

Variables	Definition
Hypotension	The systolic and diastolic blood pressure is below 90/60 mmHg
Shivering	Shaking slightly and uncontrollably during or after an anesthesia procedure
Delayed emergence	Failure to fully awake 30–60 min after anesthesia with impaired unconsciousness and breathing
Hemorrhages more than EBV	Hemorrhages more than their estimated total blood volume based on body weight and gender (for males 70 mL/kg and for females 65 mL/kg)
PONV	Nausea and vomiting occur in the first 24–48 h after surgery
Prolonged length of stay in ICU	Patients stay in ICU more than 5 days
Oxygen desaturation	Oxygen saturation below 90%
Hypothermia	Core temperature below 36.0 °C
Unexpected ventilation	Unplanned use of ventilation
Nosocomial infection	Infection that occurs within 48 h of hospitalization, three days out of the hospital, or 30 days of surgery

**Table 2 jcm-11-05292-t002:** Demographic data of geriatric patients with mortality and univariate analysis.

Variables	All Patients, *n* = 1621 *n* (%) or Mean (±)	Mortality (%)		*p*-Value
		Survival, *n* = 1551 *n* (%) or Mean (±)	Non-Survival, *n* = 70 *n* (%) or Mean (±)	
**Age, years old**	67.1 ± 6.2	66.9 ± 6.1	68.9 ± 7.1	
60–84	1603 (98.9)	1535 (95.8)	68 (4.2)	
≥85	18 (1.1)	16 (88.9)	2 (11.1)	0.18
**Gender**				
Female	743 (45.8)	713 (95.8)	30 (4.2)	
Male	878 (54.2)	838 (95.3)	40 (4.7)	0.61
**Body Mass Index, kg m^−2^**	23.0 ± 3.6	23.0 ± 3.6	22.2 ± 3.4	
Normal	1018 (62.8)	970 (95.3)	48 (4.7)	
Under/overweight	603 (37.2)	581 (96.4)	22 (3.6)	0.31
**Emergency or Nonemergency Surgery**				
Nonemergency	1495 (92.2)	1446 (96.7)	49 (3.3)	
Emergency	126 (7.8)	105 (83.3)	21 (16.7)	<0.01 *
**ASA Score**				
ASA I and II	1139 (70.3)	1107 (97.2)	32 (2.8)	
ASA III and IV	482 (29.7)	444 (92.1)	38 (7.9)	0.01 *
**Anesthesia Techniques**				
Non-General Anesthesia	527 (32.5)	514 (97.5)	13 (2.5)	
General Anesthesia	1094 (67.5)	1037 (94.8)	57 (5.2)	<0.01 *
**COVID-19 Status**				
Non-COVID	1606 (99.1)	1540 (95.8)	66 (4.2)	
COVID-19	15 (0.9)	11 (73.3)	4 (26.6)	<0.01 *
**CCI Score**				
CCI ≤ 3	1546 (95.4)	1486 (96.1)	60 (3.9)	
CCI > 3	75 (4.6)	65 (86.7)	10 (13.3)	<0.01 *
**Comorbidities**				
None	977 (60.3)	924 (94.6)	53 (5.4)	
Yes	644 (39.7)	627 (97.4)	17 (2.6)	0.01 *
- Hypertension	487 (30.0)	465 (95.5)	22 (4.5)	
- Malignancy without metastasis	219 (13.5)	208 (95.0)	11 (5.0)	
- Diabetes	211 (13.0)	199 (94.3)	12 (5.7)	
- Congestive heart failure	92 (5.7)	80 (87.0)	12 (13.0)	
- Moderate-to-severe renal disease	92 (5.7)	87 (94.6)	5 (5.4)	
- Cerebrovascular disease	62 (3.8)	54 (87.1)	8 (12.9)	
- Previous myocardial infarction	54 (3.3)	46 (85.2)	8 (14.8)	
- Peripheral vascular disease	50 (3.1)	48 (96.0)	2 (4.0)	
- Chronic pulmonary disease	33 (2.0)	30 (90.9)	3 (9.1)	
- Diabetes with chronic complications	27 (1.7)	23 (85.2)	4 (14.8)	
- Metastatic solid tumor	25 (1.5)	22 (88.0)	3 (12.0)	
- Dementia	13 (0.8)	8 (61.5)	5 (38.5)	
- Mild liver disease	10 (0.6)	9 (90.0)	1 (10.0)	
- Cerebrovascular (hemiplegia) event	9 (0.6)	8 (88.9)	1 (11.1)	
- Peptic ulcer disease	6 (0.4)	6 (100.0)	0 (0.00)	
- Moderate or severe liver disease	6 (0.4)	5 (83.3)	1 (16.7)	
- Rheumatologic disease	5 (0.3)	4 (80.0)	1 (20.0)	
- Lymphoma	3 (0.2)	3 (100.0)	0 (0.0)	
- Leukemia	2 (0.1)	2 (100.0)	0 (0.0)	
- AIDS	0 (0.0)	-	-	
**Intraoperative AE**				
None	1321 (81.5)	1276 (96.6)	45 (3.4)	
Yes	300 (18.5)	275 (91.6)	25 (8.4)	<0.01 *
- Hypotension	215 (12.7)	196 (91.2)	19 (8.8)	
- Hemorrhages more than EBV	58 (3.4)	53 (91.4)	5 (8.6)	
- Prolonged surgery time	50 (2.9)	44 (88.0)	6 (12.0)	
- Oxygen desaturation	11 (0.6)	8 (72.7)	3 (27.3)	
- Anesthesia awareness	3 (0.2)	2 (66.7)	1 (3.3)	
- Anesthesia technique conversion	2 (0.2)	2 (100.0)	0 (0.0)	
- Prolonged block	1 (0.1)	1 (100.0)	0 (0.0)	
- Others	29 (1.7)	27 (92.1)	2 (6.9)	
**AE in the recovery room**				
None	1485 (91.6)	1429 (96.2)	56 (3.8)	
Yes	136 (8.4)	122 (89.7)	14 (10.3)	<0.01 *
- Hypotension	67 (4.1)	58 (86.6)	9 (13.4)	
- Shivering	34 (2.1)	34 (100.0)	0 (0.0)	
- Delayed emergence	17 (1.0)	14 (82.4)	3 (17.6)	
- Hemorrhages more than EBV	14 (0.8)	10 (71.4)	4 (28.6)	
- PONV	10 (0.6)	9 (90.0)	1 (10.0)	
- Pain	7 (0.4)	7 (100.0)	0 (0.0)	
- Oxygen desaturation	5 (0.3)	3 (60.0)	2 (40.0)	
- Hypothermia	3 (0.2)	3 (100.0)	0 (0.0)	
**Postoperative ICU care**				
No	1403 (86.6)	1364 (97.2)	39 (2.8)	
Yes	218 (13.4)	187 (85.8)	31 (14.2)	<0.01 *
**AE in the ICU**				
None	150 (68.8)	135 (90.0)	15 (10.0)	
Yes	68 (31.2)	52 (76.5)	16 (23.5)	<0.01 *
- Prolonged length of stay in ICU	39 (17.2)	30 (76.9)	9 (23.1)	
- Unexpected ventilation used	18 (7.9)	14 (77.8)	4 (22.2)	
- Nosocomial infection	6 (2.6)	2 (33.3)	4 (66.7)	
- Others	14 (6.2)	11 (78.6)	3 (21.4)	
**AE in the inward**				
None	1511 (93.2)	1452 (96.1)	59 (3.9)	
Yes	110 (6.8)	99 (90.0)	11 (10.0)	<0.01 *
- Pain	60 (3.7)	59 (98.3)	1 (1.7)	
- Hypotension	19 (1.2)	11 (57.9)	8 (42.1)	
- PONV	14 (0.9)	14 (100.0)	0 (0.0)	
- Hemorrhage	7 (0.4)	7 (100.0)	0 (0.0)	
- Oxygen desaturation	5 (0.3)	1 (20.0)	4 (80.0)	
- Others	23 (1.5)	19 (82.6)	4 (17.4)	

* *p* < 0.05. Abbreviations: AE, adverse event; AIDS, acquired immune deficiency syndrome; ASA, American Society of Anesthesiologists; CCI, Charlson Comorbidity Index; COVID-19, coronavirus; EBV, estimated blood volume; ICU, intensive care unit; PONV, Postoperative Nausea and Vomiting.

**Table 3 jcm-11-05292-t003:** Univariate and multivariable logistic regression analysis for patient characteristics with mortality of geriatric patients undergoing surgery.

Variables	Mortality (%)		Univariate		Multivariable	
	Survival	Non-Survival	OR (95% CI)	*p*-Value	OR (95% CI)	*p*-Value
Age 85 y.o	16 (88.9)	2 (11.1)	2.82 (0.64–12.52)	0.18	-	-
Male	838 (95.3)	40 (4.7)	1.13 (0.70–1.84)	0.61	-	-
BMI Under/overweight	581 (96.4)	22 (3.6)	0.77 (0.46–1.28)	0.31	-	-
COVID-19	11 (73.3)	4 (26.6)	8.49 (2.63–27.35)	<0.01 *	4.34 (1.04–18.07)	0.04 *
Comorbidities	627 (97.4)	17 (2.6)	0.47 (0.27–0.82)	0.01 *	0.73 (0.38–1.38)	0.33
CCI > 3	65 (86.7)	10 (14.30)	3.81 (1.87–7.78)	<0.01 *	2.33 (1.03–5.26)	0.04 *
Emergency surgery	105 (83.3)	21 (16.7)	5.90 (3.41–10.21)	<0.01 *	3.70 (1.96–7.00)	<0.01 *
General Anesthesia	1037 (66.90)	57 (81.40)	2.96 (1.83–4.80)	<0.01 *	1.85 (0.96–3.59)	0.07
ASA III and IV	444 (28.60)	38 (54.30)	2.17 (1.180–4.01)	0.01 *	1.44 (0.79–2.63)	0.20
Intraoperative AE	275 (91.6)	25 (8.4)	2.58 (1.55–4.28)	<0.01 *	0.94 (0.45–1.96)	0.86
AE in the recovery room	122 (89.7)	14 (10.3)	2.93 (1.59–5.41)	<0.01 *	0.64 (0.26–1.59)	0.34
Postoperative ICU care	187 (85.8)	31 (14.2)	5.50 (3.35–9.05)	<0.01 *	2.70 (1.32–5.53)	0.01 *
AE in the ICU	52 (76.5)	16 (23.5)	8.54 (4.58–15.92)	<0.01 *	3.43 (1.32–8.96)	0.01 *
AE in the inward	99 (90.0)	11 (10.0)	2.34 (1.39–5.37)	<0.01 *	0.46 (0.17–1.24)	0.12

* *p* < 0.05. Abbreviations: AE, adverse event; ASA, American Society of Anesthesiologists; BMI, Body Mass Index; CI, confidence interval; CCI, Charlson Comorbidity Index; COVID-19, coronavirus; ICU, intensive care unit; OR, odds ratio; y.o, years old.

## Data Availability

Not applicable.

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
