# Peer review of "Perioperative Factors Impact on Mortality and Survival Rate of Geriatric Patients Undergoing Surgery in the COVID-19 Pandemic: A Prospective Cohort Study in Indonesia"

_jcm, 2022, doi:10.3390/jcm11185292_

Round 1

Reviewer 1 Report

I appreciate the opportunity to review your manuscript. The authors have conducted a retrospective cohort study investigating the perioperative factors impact on mortality and survival rate of geriatric patients receiving anesthesia services in COVID-19 pandemic era. This study is to determine the association between perioperative factors with 30-days mortality and survival rate of geriatric patients who receive anesthesia services in COVID-19 pandemic.

Overall, most parts of the manuscript are well-written and interesting, but this study need some corrections, and the authors need to address the following comments to the reviewer.

Major points

1.     Abstract. “The total patients that survive within 30 days of observation were 4.3%. COVID-19, CCI>3, emergency surgery, postoperative ICU care, and adverse event in the ICU were significantly associated factors with p<0.05.”

-I suspected this logistic regression analysis was overfitted. The number of primary outcome  (30 days mortality) was 70. It has been suggested that the data should contain at least ten events for each variable entered into a logistic regression model [1].  Therefore, in general, only 7 factors (70/10) can be included and examined. More factors leads to overfitting results. In addition, the author should not emphasize only P<0.05, more detail description of results needed (estimate, 95% confidence interval, and P value)[2].

The authors should discuss more about this problem. The following papers may be helpful:

[1] Peduzzi, P.; Concato, J.; Kemper, E.; Holford, T.R.; Feinstein, A.R. A Simulation Study of the Number of Events per Variable in Logistic Regression Analysis. J Clin Epidemiol 1996, 49, 1373–1379, doi:10.1016/s0895-4356(96)00236-3.

[2] https://www.tandfonline.com/doi/full/10.1080/00031305.2019.1583913

2.     I wonder the 30 day mortality of pre-pandemic era in Indonesia. This information can provide valuable information to compare the clinical outcomes between pre-pandemic and pandemic-period.

3.     Is Kaplan-Meier curve necessary? The author discusses only 30 day mortality and not long-term outcome. In addition, I think this prospective study is hypothesis-generating study and not confirmatory study. Thus, survival analysis is unnecessary for this kind study.

Minor points

1.     univariate/multivariate, logistic regression

- In the strict sense, multivariate analysis refers to simultaneously predicting multiple outcomes. Since the author deals with techniques that use multiple variables to predict a single outcome (30 day mortality), I recommend the more general term “multivariable” analysis.

2.     P2L81: AE

Please add abbreviation when “AE” appeared firstly.

3.     P3L119–: ”Intraoperative factors included adverse events (AE) that occur in the operating room, such as hypotension, hemorrhage more than Estimated Blood Volume (EBV), prolonged surgery time, oxygen saturation, anesthesia awareness, anesthesia technique conversion, and prolonged block”.

-As this prospective cohort study is multicenter study, more definite definition of each adverse events are needed for internal validity. The author should describe more details of each definition of AEs. For example, how is the “oxygen desaturation” in intraoperative SE?

4.     Table 1.

-In general,(%)is for N.

In Table 1, the author describe (%) in line.

For example, in survival (N=1551), female 713 (95.8) and male 838 (95.3).
This can confuse the readers.  

Please indicate whether is the (%) is line or row to avoid misunderstanding.

Reviewer 2 Report

The study is aimed to determine the association between perioperative factors with 30-days mortality and survival rate of geriatric patients who receive anesthesia services during the COVID-19 pandemic.    The title is “Perioperative factors impact on mortality and survival rate of geriatric patients receiving anesthesia services in COVID-19 pandemic: A prospective cohort study in Indonesia”.

1.        This is a prospective cohort study.

2.        Several factors influence the outcome of the study.  Please discuss these.

3.     How about these perioperative factors in western countries?  What are the differences?

4.        What are the differences between the COVID-19 pandemic and the non-COVID-19 period?

5.        Please review the literature and add more details in the discussion section.

6.        What is the new knowledge of the study?

Please recommend to the readers “How to apply this knowledge?”.

Round 2

Reviewer 1 Report

No specific comments for the revised manuscript.